# Validation of a culturally adapted Swedish-language version of the Death Literacy Index

Therese Johansson [1,2]*, Åsa Olsson[1,3], Carol Tishelman [4,5,6], Kerrie Noonan [7,8,9], Rosemary Leonard[7], Lars E. Eriksson [1,10,11], Ida Goliath[1,12☼], Joachim Cohen[6☼]

1 Department of Neurobiology, Care Sciences and Society, Karolinska Institutet, Huddinge, Sweden, 2 Cicely Saunders Institute of Palliative Care, Policy & Rehabilitation, King's College London, London, United Kingdom, 3 Swedish National Graduate School on Ageing and Health (SWEAH), Lund University, Lund, Sweden, 4 Department of Learning, Informatics, Management and Ethics, Karolinska Institutet, Solna, Sweden, 5 Stockholm Health Care Services, Region Stockholm, Sweden, 6 End-of-Life Care Research Group, Vrije Universiteit Brussel and Ghent University, Brussels, Belgium, 7 School of Social Sciences, Western Sydney University, Sydney, Australia, 8 Death Literacy Institute, Australia, 9 Public Health Palliative Care Unit, La Trobe University, Melbourne, Australia, 10 School of Health and Psychological Sciences, City, University of London, London, United Kingdom, 11 Medical Unit Infectious Diseases, Karolinska University Hospital, Huddinge, Sweden, 12 Stockholm Gerontology Research Center, Stockholm, Sweden

☼ These authors contributed equally to this work.
* therese.johansson@ki.se

**Data Availability Statement:** Data contain potentially sensitive personal information and cannot be shared publicly because of limitations in the ethical approval. We are unable to share data

## Abstract

The death literacy index (DLI) was developed in Australia to measure death literacy, a set of experience-based knowledge needed to understand and act on end-of-life (EOL) care options but has not yet been validated outside its original context. The aim of this study was to develop a culturally adapted Swedish-language version of the DLI, the DLI-S, and assess sources of evidence for its validity in a Swedish context. The study involved a multi-step process of translation and cultural adaptation and two validation phases: examining first content and response process validity through expert review ($n = 10$) and cognitive interviews ($n = 10$); and second, internal structure validity of DLI-S data collected from an online cross-sectional survey ($n = 503$). The psychometric evaluation involved analysis of descriptive statistics on item and scale-level, internal consistency and test-retest reliability, and confirmatory factor analysis. During translation and adaptation, changes were made to adjust items to the Swedish context. Additional adjustments were made following findings from the expert review and cognitive interviews. The content validity index exceeded recommended thresholds (S-CVI$_{Ave}$ = 0.926). The psychometric evaluation provided support for DLI-S' validity. The hypothesized six-factor model showed good fit ($\chi^2 = 1107.631$ $p<0.001$, CFI = 0.993, TLI = 0.993, RMSEA = 0.064, SRMR = 0.054). High internal consistency reliability was demonstrated for the overall scale (Cronbach's $\alpha = 0.94$) and each sub-scale ($\alpha$ 0.81–0.92). Test-retest reliability was acceptable, ICC ranging between 0.66–0.85. Through a comprehensive assessment of several sources of evidence, we show that the DLI-S demonstrates satisfactory validity and acceptability to measure death literacy in the Swedish context. There are, however, indications that the sub-scales measuring community capacity perform worse in comparison to other sca and may function differently in Sweden than in the original context. The DLI-S has potential to contribute to research on community-based EOL interventions.

sets due to GDPR restrictions in Sweden and the EU. Data are available upon reasonable request from prefekt@nvs.ki.se.

**Funding:** "This study was funded by Strategic Research Area Health Care Science (SFO-V), Karolinska Institutet; the Doctoral School in Health Care Sciences, Karolinska Institutet; Forskningsrådet om Hälsa, Arbetsliv och Välfärd (Swedish Research Council for Health, Welfare and Working Life, grant number 2014-04071); Stockholm City Elder Care Bureau; and Stockholm Gerontology Research Center. The funders had no part in, nor influence on, the study design, data collection, analysis, interpretation, or writing of results.".

**Competing interests:** The authors have declared that no competing interests exist.

# Introduction and aim

Just as health literacy relates to the extent to which people can find, interpret, and use health information and services [1], death literacy (DL) is a newly-coined term denoting knowledge and skills needed to understand and act on options for end-of-life (EOL) and death care [2]. The concept of DL was developed after extensive qualitative research involving carers and networks involved in caring for a person dying at home. Building on new public health approaches that highlight individual and community capacity-building and empowerment to handle issues of dying, death, and loss [3], DL is theorized to develop from engaging with EOL care and learning from those experiences. It is defined as *"a set of knowledge and skills that make it possible to gain access to, understand, and act upon end-of-life and death care options"* [4].

The *death literacy index* (DLI), an operationalization of DL, is a questionnaire developed from research in Australia about people's experiences of EOL care [5]. The DLI evolved through an iterative mixed-methods approach to content development, refinement, and testing, and is intended to measure group levels of DL and evaluate EOL-related educational initiatives [6]. As such, the DLI may constitute a promising tool for appraising impact of health promoting activities in relation to EOL issues. To date, the DLI has primarily been used among adults in the Australian general public, although the questionnaire has garnered increasing interest internationally, and been validated in the United Kingdom [7], Turkey [8], and China [9]. However, the extent to which DL and its operationalization are comparable across cultures is unknown. The aim of this study was to develop and culturally adapt Swedish-language version of the DLI (DLI-S) and assess its validity in a Swedish context.

# Materials and methods

## Study design

This validation study [10] used a multi-step approach combining qualitative and quantitative methods [11] to translate and culturally adapt the Death Literacy Index (DLI) into Swedish and to assess the validity of the resulting DLI-S [12]. The multi-step process comprised: forward translation and adaptation by initial translators; translation revision through negotiated consensus in the research team; review by a panel of experts; pre-testing using cognitive interviews; literacy reviews by an external consultant; and psychometric evaluation using online testing of the DLI-S with a survey panel (see Fig 1).

## The Death Literacy Index

The DLI is a multi-dimensional self-report questionnaire containing 29 items distributed over four dimensions of DL; *Practical knowing* (n of items = 8), *Learning from Experience* (n = 5), *Factual knowledge* (n = 7), and *Community capacity* (n = 9) [2]. The dimensions are represented in the DLI as four scales, two of which contain sub-scales capturing specific dimensional facets; *Practical knowing* has the sub-scales *Talking support* (n = 4) and *Hands-on care* (n = 4), whereas *Community Capacity* has the sub-scales *Accessing help* (n = 5) and *Community support groups* (n = 4). DLI items are in the form of statements with ordered category responses on a 5-point Likert-type scale, usually ranging from "do not agree at all" and "strongly agree". Since the DLI covers various aspects of DL, scores are calculated for each scale and sub-scale, using transformed mean scores. Total DL is calculated as a higher-order composite score, which is said to reflect overall capacity gained from previous experiences [6]. The proposed six-factor model in the original English-language DLI demonstrated satisfactory psychometric properties in Australia indicating good model fit overall (CFI = 0.955,

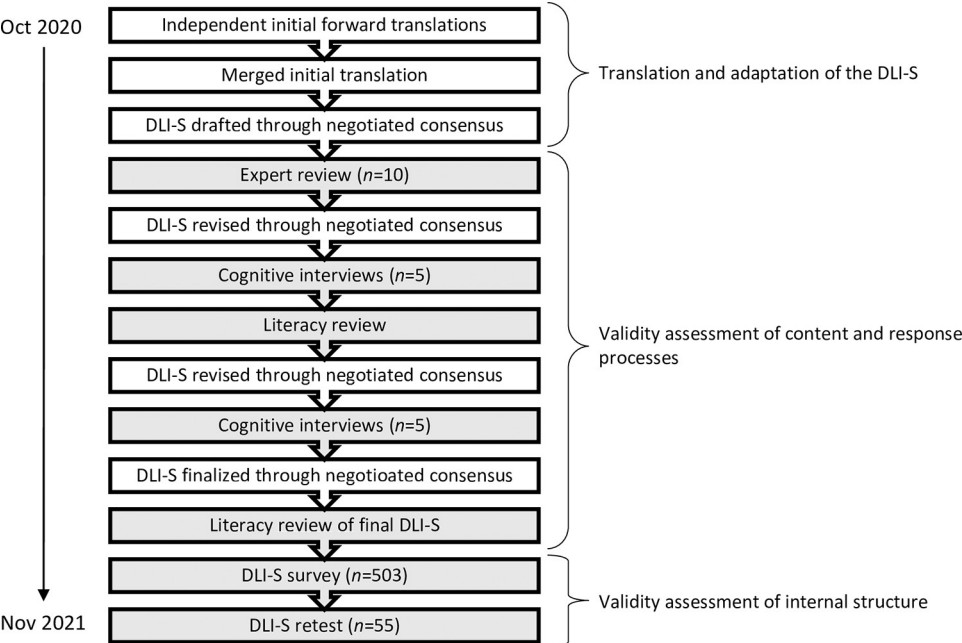

**Fig 1. Schematic of the study process.** Steps in the translation and adaptation of the DLI-S. Sources of validity evidence.

TLI = 0.95, RMSEA = 0.049), and high reliability of the scales and sub-scales (Cronbach's $\alpha$ ranging 0.82 to 0.95) [2], and the United Kingdom (CFI = 0.94, RMSEA = 0.07, SRMR = 0.07; Cronbach's α for sub-scales ranging 0.76 to 0.93) [7]. Moreover, translated DLI versions have demonstrated acceptable reliability and validity in Turkey (total CVI = 0.91; CFI = 0.93, RMSEA = 0.053; Cronbach's α ranging 0.68–0.90) [8] and China (CFI = 0.956, RMSEA = 0.055, SRMR = 0.051, Cronbach's α for sub-scales ranging 0.76–0.97) [9].

## Translation and adaptation of the Swedish Death Literacy Index

After obtaining permission to translate the DLI from the original developers, co-authors RL and KN, the DLI-S development process was guided by recommendations for translation and cross-cultural adaptation [13, 14]. Back translation was not used in this study as it has been criticized for missing variation in linguistic meaning and cultural nuances [11, 15, 16]. The 29 items with corresponding instructions and response categories were independently translated from English to Swedish by co-authors TJ and ÅO, both native Swedish speakers proficient in English. The two initial forward translations were compared to identify discrepancies and discuss ambiguous wordings, resulting in a joint draft. This DLI-S draft was sent to the members of a multidisciplinary research group with both native English and Swedish speakers, who individually reviewed the draft, prior to continued revision through a process of negotiated consensus in a series of meetings [11] after which changes to content were decided so that the Swedish items would convey the same conceptual understanding as the original DLI [14, 17]. The original DLI and items in the DLI-S are presented in S1 and S2 Appendices respectively.

## Validation

Validity is defined as a unitary concept derived from several sources of evidence that each contribute to the overall validity of a measure [18]. Validation involves the assessment of evidence

to support interpretations of scores in relation to the intended use of a measure [19]. Validity is context-specific, and thus, validation is strongly recommended whenever a measure is to be used in a new, qualitatively different, population or context [20]. In this article we focus on three sources of validity evidence: 1) *Evidence based on content*, relating to the adequacy and relevance of items to represent and score the construct measured [21]; 2) *Evidence based on response processes*, which involves exploration of respondents' actions and cognitive processes to identify possible sources of error, e.g., challenges with interpreting and answering items [18]; and 3) *Evidence based on internal structure*, which concerns how items relate to each other and to the overarching construct, often measured as reliability across items, time, or respondents [21]. These correspond with identified quality criteria for assessing psychometric properties of measures [12].

## Data collection

**Validity evidence based on content and response process.**     *Expert panel review*. An expert review was conducted to evaluate the validity of DLI-S items to measure DL [22]. Since DLI targets the general population, there is no delimited area of expertise relevant for this step. We therefore made efforts to recruit ten panel members with varying ages, backgrounds, and personal and professional perspectives in relation to the EOL, e.g., palliative care, gerontology, ethnology, professional translation, clinical nursing, and from patient interest organizations. Email invitations were sent with information about the study purpose and methods. Inclusion criteria were proficiency in both Swedish and English. Each panel member reviewed the DLI-S independently, using an online survey, accessible only after providing informed consent. Following recommendations by Grant and Davis [22], panel members were first provided with a summary description of the conceptual DL model and provided with information about the intended use of the DLI. As shown in S3 Appendix, the review comprised two main sections: *Translation review*, in which the semantic and cultural equivalence between each Swedish item and the original corresponding English-language item was assessed on a four-point scale; and *Content validity assessment*, in which the DLI-S items' relevance and clarity of content were rated on a four-point scale. Panel members could comment and suggest changes for every item throughout both sections of the survey.

*Pre-testing using cognitive interviews*. To determine whether the DLI-S items were acceptable, comprehensible, and able to generate information as intended, authors TJ and ÅO conducted cognitive interviews with a new sample from the target population [23], i.e., adults from the general public. Interviews combined think-aloud technique, in which the respondent describes their reasoning out loud as they read and respond to each item [24], and verbal probing, whereby the interviewer asks questions to clarify and further explore any issues [25]. Participants were recruited through convenience sampling in the researchers' networks. No explicit inclusion or exclusion criteria were used but we strove for variation of age, gender, educational level, and professional background. Due to the Covid-19 pandemic, interviews were conducted online using Zoom. All participants received written information about the study in advance and consented orally to participate in audio-recorded interviews. While we assessed the risk of harm to participants as low, the interviewers were experienced in handling emotional reactions and could refer participants to other sources for support if needed.

*Literacy review*. To investigate linguistic accessibility, the lexical profile of the DLI-S was reviewed by an independent consultant on two occasions: after the first 5 cognitive interviews and again after the DLI-S had been finalized (Fig 1).

**Validity evidence based on internal structure.**     *Online testing with survey panel*. Data was collected from September–November 2021 through an online survey administered by an

external data collection agency with a pre-existing panel of ca 100,000 Swedish residents, aged 15 or older, willing to partake in surveys on various topics for both market and scientific research.

Since this study aimed for theoretical generalization rather than making statistical inferences regarding population estimates of death literacy, a representative probability sample was not necessary [26]. Still, we strove for a sample reflective of the heterogeneity within the Swedish population. The inclusion criteria were being eighteen years old or over and residing in Sweden. No explicit exclusion criteria were used and no EOL experience was required to participate. Survey invitations were sent to a quota sample ($n$ = 2991) in the agency's existing panel, stratified by gender, age, and geographical region. The minimum sample size was set to 500 to have sufficient data and power for confirmatory factor analysis (CFA) [27]. The survey comprised the DLI-S items and questions about sociodemographic variables and EOL experiences and was only accessible to panel members who agreed to participate after being informed about the study. Participants were notified that they could exit the survey at any time and were provided with the researchers' contact information in case they needed support or had questions or comments following participation. At the end of the survey, participants were asked whether they were willing to complete the survey a second time, to assess test-retest reliability. Of the 412 that agreed to participate in a follow-up, 82 were re-invited to the survey. The retest sample size was set to minimum 50 to provide sufficient for calculating intra-class correlation coefficients (ICC) [28]. The time interval was set to approximately 4 weeks, chosen to allow enough time to avoid rehearsal effects but short enough to minimize the risk of participants experiencing real change that might alter their responses [29].

## Data analysis

**Expert panel review.**   Comments related to the translation and content of each item were reviewed and summarized by TJ. Quantitative ratings were collated in a matrix in Microsoft Excel to calculate the content validity index (CVI), i.e., inter-rater agreement at the item-level (I-CVI) and scale-level (S-CVI). I-CVI represents the proportion of panel members rating an item positively (e.g., 3 or 4) and is recommended to be at least 0.78 [30]. S-CVI was calculated using average proportion, recommended for panels of $\geq 8$ [31].

**Pre-testing using cognitive interviews.**   Data were based on the interviewers' field notes, using audio-recordings as back-up if needed, and compiled into a summary matrix, linking comments to the items and sub-scales to which they referred [32]. The matrix was reviewed in recurring consensus meetings to inform decisions regarding item retention, revision, or deletion and modifications to instructions or response categories.

**Literacy review.**   The literacy review was conducted using the software *AntWordProfiler* to examine the proportion of words among the 5,000 most common in Swedish.

**Online testing with survey panel.**   Statistical analyses were conducted using IBM SPSS Statistics 28 and the *lavaan* package [33] in R (version 4.1.1). Descriptive statistics were used to explore socio-demographic characteristics of the sample and analyze central tendencies and dispersion on item- and scale-level. Response variation was examined at item-level by investigating if all five response categories were used and at scale-level by identifying whether there were floor or ceiling effects, i.e., $\geq 15\%$ of responses in the maximum and/or minimum category [34]. Inter-item correlations were calculated for all items.

Corrected item-total correlations were examined and values between 0.2–0.7 were considered good discrimination within a scale [35]. Internal consistency reliability was assessed by calculating Cronbach's α coefficients and confidence intervals of items, sub-scales and the full DLI. Values $\geq 0.8$ were considered as demonstrating good reliability [36]. However, since the

appropriateness of Cronbach's α as a sole measure of reliability has been questioned, due to the effects of the number of scale items and assumptions of unidimensionality and tau equivalence [37, 38], we also calculated average inter-item correlation (AIC), a more robust indicator of internal consistency [39]. The acceptable range for AIC is considered to be 0.15–0.5 [36]. ICC was calculated to assess scale test-retest reliability, using the two-way mixed effects, single measure model (ICC type 3,1). ICC >0.9 indicates excellent reliability, whereas ICC >0.75 is considered good, >0.5 is moderate and <0.5 suggests poor reliability [40].

Since a hypothesized model had been identified and tested during the development of the original DLI and the current study sought to validate this, a CFA was considered more appropriate to assess the fitness of the DLI factor structure than an exploratory factor analysis [41, 42]. Three factor models were tested: with DLI treated as one universal factor; as four factors (corresponding to the original main DL dimensions); and as six factors (corresponding to four sub-scales (*1.1*, *1.2*, *4.1*, *4.2*) and two dimension-scales (*2* and *3*). As the 5-point response range used in the DLI was relatively short, data were modelled as ordinal rather than continuous. Diagonally weighted least squares (DWLS) were used as model estimator, since this has been shown to perform better than robust maximum likelihood estimation with ordinal data [41]. Tucker-Lewis index (TLI) and Comparative Fit Index (CFI) >0.95, root mean square error of approximation (RMSEA) >0.06, Standardized Root Mean Square Residual (SRMR) >.08, and $\chi^2$ $p$ value >.05 were used as indices of good model fit [43].

## Ethical considerations of the study

The study process described below was approved by the Swedish Ethics Review Authority (reference number 2021–00915) and conducted according to the ethical principles of the Helsinki Declaration [44]. All study participants received written information about the nature of the study, including its subject, purpose, and procedure, as well as their right to withdraw. Informed consent to participate was obtained from all participants. The DLI asks about experiences related to the EOL, a potentially sensitive topic. However, previous research has shown that questions addressing dying or death may have an effect on immediate mood but unlikely to cause harm [45]. Furthermore, it should be noted that participants were not persons known to be at the EOL themselves.

## Results

### Translation and adaptation of the DLI-S

Minor changes were made to item content to ensure conceptual equivalence to the original items and better adapt items to the Swedish context. Detailed examples of changes made at various stages of the item revision process during both translation/adaptation and validation phases are provided in S1 Table, using two DLI-S items as examples.

### Validity based on content and response process

**Expert panel review.** All ten individuals contacted agreed to participate, of whom seven were women. The qualitative data, e.g., potential concerns identified and suggestions for improvement, were addressed in discussions about item revision during consensus meetings with the research group, thereby informing the continued process of revising the DLI-S. Overall, the expert review identified words and terms that were awkward or unclear in Swedish and raised questions regarding content relevance in Sweden. To address these issues, minor changes to content were made for several items (see S1 Table). I-CVI scores ranged between 0.837–0.987, indicating that each item was considered relevant for the DL construct. The full

DLI-S demonstrated good content validity with an S-CVI$_{Ave}$ = 0.926. All CVI scores are presented in S2 Table.

**Cognitive interviews.** In total, ten people (seven women) participated in the cognitive interviews. Overall, the cognitive interviews showed that some items could raise memories of past experiences. Nevertheless, DLI-S content was generally acceptable to participants, i.e., not distressing, interesting, and with items of varying difficulty. Table 1 presents six types of issues requiring minor changes to the DLI-S and affected items and/or scales, based on participants' response processes, comments, and suggestions for improvement (see S1 Table for more detailed examples of item revision). For example, we found that participants' responses to the question in sub-scale *Talking Support* (1.1), "*how difficult or easy you would find the following* [items]", indicated that the question did not sufficiently prompt participants to think about their self-perceived competence or preparedness for engaging in conversations about EOL issues when answering, as was intended. Instead, they often mentioned a combination of values, perceived social taboos, or relation to the conversation partner when thinking aloud about their responses to the items. Consequently, the question was reformulated to "*how prepared would you be to talk about the following* [conversations about EOL issues]?", to better guide respondents to consider their readiness to engage in conversations about the EOL when answering items in this sub-scale.

**Literacy review.** The first review found that 84.7% of the words used in the DLI-S were among the 5,000 most common in Swedish. The consultant suggested more common or easy-to-read alternatives to difficult or uncommon terms, which formed the basis for another round of item revision through consensus meetings in the research group. This revised DLI-S version was then used in the five subsequent cognitive interviews. A second literacy review was conducted on the final DLI-S, in which the rate had increased to 93.3%, which was deemed sufficient.

## Validity based on internal structure

In total, 503 people completed the survey, giving a response rate of 17%. At the retest, 55 participants completed the survey, giving a second response rate of 67%. Socio-demographic

**Table 1. Overview of issues identified in the cognitive interviews and how they were addressed.**

| Issue | Affected items/ scales | Revisions to address issue |
|---|---|---|
| Vague or ambiguous item statement | 4, 13, 14, 16,19, 24 | Items clarified to better reflect conceptual meaning and/or semantic precision |
| Double-barrelled item, e.g., item statement contains more than one behaviour/trait | 19 | First section removed to make item statement more concise and focused on the trait in question |
| Item relevance or applicability to Swedish context | 14, 15, 20, Scale 4.1 | Examples provided clarify content (21); item wordings revised and refined (14, 15, 21). Question in scale 4.1 rephrased to better suit the Swedish context with universal health care |
| Overlapping items (item content is perceived as repeating) | Scale 2 | No change at this stage, all items retained as they are considered to contribute with different aspects on a theoretical level |
| Unclear question format | Sub-scales 1.1 and 4.1 | Questions specified to better reflect intended use of the DLI; clarifying instructions added to highlight that questions relate to individual perceptions and that there are no right or wrong answers |
| Unsuitable or unclear response categories | Sub-scales 1.1 and 1.2 | Response categories changed to better suit the scale question (from *very difficult/very easy* to *not prepared at all/very prepared*) |

characteristics of participants are presented in Table 2. Since the online survey used a mandatory response procedure, requiring all items to be answered to proceed, there were no missing values in the data. Item-level descriptives are presented in Table 3. Inter-item correlations within sub-scales were generally high, e.g. >0.5, and presented in S3 Table.

**Scale descriptives.** Scale descriptives, i.e., mean scores, standard deviations, and floor and/or ceiling effects, are presented in Table 4. Total DLI scores and sub-scale scores were normally distributed in the sample, except for the sub-scale *Accessing help*, which demonstrated a floor effect, i.e., a negatively skewed distribution.

**Reliability.** The DLI-S demonstrated high internal consistency reliability, Cronbach's $\alpha$ = 0.94 for the overall scale and between 0.81–0.92 for the sub-scales. Test-retest reliability was moderate to good, with scale-level ICC ranging 0.66–0.84. Reliability estimates for the full DLI-S and each sub-scale are presented in Table 4.

**Table 2. Sample characteristics of participants in Phase 2 (*n* = 503).**

| Socio-demographic characteristics | Mean (SD) | Range |
|---|---|---|
| **Age** | **49.95 (17.92)** | **18–86** |
| | Count | Percentage |
| **Gender** | | |
| Male | 253 | 50.4% |
| Female | 246 | 49.0% |
| Other (Non-binary or trans) | 3 | 0.6% |
| **Highest level of completed education** | | |
| Lower secondary education or less | 42 | 8.3% |
| Upper secondary education | 207 | 41.2% |
| Post-secondary education | 24 | 4.8% |
| Higher general or vocational education diploma | 77 | 15.3% |
| Higher education, bachelor's degree or equivalent | 89 | 17.7% |
| Higher education, master's degree or more | 64 | 12.7% |
| **Origin** | | |
| Sweden | 467 | 92.8% |
| Europe, excl. Sweden | 26 | 5.2% |
| Outside Europe | 10 | 2.0% |
| **Work or volunteering experience** | | |
| EOL care provision (work / volunteer) | 71 / 28 | 14% / 5.6% |
| Grief support (work / volunteer) | 49 / 32 | 9.9% / 6.2% |
| **Professional experience in care** | | |
| Care sector | 60 | 11.9% |
| Social care sector | 39 | 7.8% |
| Both care and social care | 12 | 2.4% |
| No professional experience | 392 | 77.9% |
| **EOL experiences** [a] | | |
| Death of a family member, close relative, or friend | 408 | 81.2% |
| Own life-threatening illness | 45 | 9.0% |
| Supporting a person with a life-threatening illness | 113 | 22.4% |
| Care for a relative at the EOL | 64 | 12.8% |
| Providing EOL care professionally | 55 | 10.9% |
| No EOL experience | 29 | 5.8% |

Notes: EOL = end-of-life; [a] Multiple responses allowed, sum≠503

**Table 3. Descriptive statistics for the DLI-S items\* (*n* = 503).**

| Items | Mean | SD | Corrected item-total correlation |
|---|---|---|---|
| 1. Talk about dying, death, or grief with a close friend | 3.91 | 1.04 | .43 |
| 2. Talk about dying, death, or grief with a child | 2.98 | 1.27 | .40 |
| 3. Talk with a bereaved person about their loss | 3.58 | 1.11 | .53 |
| 4. Talk with care staff about support for a person who will die at home or in their place of care | 3.66 | 1.12 | .61 |
| 5. Feed or help someone to eat | 3.75 | 1.21 | .51 |
| 6. Wash someone | 3.09 | 1.39 | .55 |
| 7. Lift someone or help to move them | 3.64 | 1.27 | .38 |
| 8. Administer injections | 2.55 | 1.51 | .46 |
| 9. Made me more emotionally prepared to support others with processes related to death and dying | 3.56 | 1.06 | .57 |
| 10. Made me think about what is important and not important in life | 3.98 | 0.99 | .43 |
| 11. Made me wiser and given me new understanding | 3.70 | 0.96 | .49 |
| 12. Increased my compassion toward myself | 3.38 | 1.05 | .38 |
| 13. Made me better prepared to face similar challenges in the future | 3.60 | 1.02 | .58 |
| 14. I know about rules and regulations regarding deaths at home | 2.12 | 1.18 | .66 |
| 15. I know that there are documents that can help a person plan before death | 3.28 | 1.38 | .47 |
| 16. I know enough about how [the health and social care systems] operate to be able to support a person in receiving care at the end of life | 2.60 | 1.29 | .75 |
| 17. I know about processes for funerals, where I can turn, and which choices are available | 3.26 | 1.31 | .58 |
| 18. I know how to access palliative care in the area where I live | 2.27 | 1.33 | .69 |
| 19. I know enough to make decisions about medical treatments and understand how they may affect quality of life, at the end of life | 2.53 | 1.33 | .70 |
| 20. I am aware of different ways that cemetery staff can be of help around funerals | 2.67 | 1.25 | .64 |
| 21. To get support in the area where I live, e.g., from clubs, associations, or volunteer organizations | 2.12 | 1.16 | .68 |
| 22. To get help with providing day to day care for a person at the end of life | 2.48 | 1.31 | .74 |
| 23. To get equipment that are required for care | 2.69 | 1.36 | .69 |
| 24. To get support that is culturally appropriate for a person | 2.06 | 1.15 | .63 |
| 25. To get emotional support for myself | 2.46 | 1.25 | .72 |
| 26. People with diseases that might lead to death | 2.83 | 1.28 | .66 |
| 27. People who are nearing the end of their lives | 2.61 | 1.28 | .70 |
| 28. People who are caring for someone who is dying | 2.50 | 1.29 | .72 |
| 29. People who are grieving | 2.86 | 1.32 | .64 |

\* DLI-S items translated to English

**Confirmatory factor analysis.** Fit indicators for all tested models are presented in Table 5. The one-factor model was tested first, demonstrating adequate fit. The four-factor model, corresponding to the main dimensions of DL, generated a better fit, though still not meeting all recommended thresholds. The six-factor model showed good fit, performing best of the three models tested. Factor loadings in this model were generally high, ranging between 0.57–0.95, and correlations between factors were moderate to strong, between 0.40–0.86 (Fig 2).

**Table 4. Descriptive statistics and reliability estimates for the full DLI-S and each sub-scale.**

|  | Mean score (SD)[a] | Floor/ceiling effect (%) | Cronbach's α (95% CI)[b] | AIC[b] | ICC (95% CI)[c] |
|---|---|---|---|---|---|
| DLI-S (full scale) | 5.15 (1.86) | 0.2/0.6 | 0.94 (0.94–0.95) | 0.36 | 0.85 (0.76–0.91) |
| Talking support | 6.28 (2.28) | 0.8/8.9 | 0.82 (0.78–0.84) | 0.52 | 0.68 (0.50–0.80) |
| Hands-on care | 5.63 (2.69) | 1.6/10.3 | 0.81 (0.78–0.84) | 0.53 | 0.81 (0.69–0.88) |
| Learning from experience | 6.59 (2.05) | 0.2/8.7 | 0.83 (0.84–0.88) | 0.56 | 0.66 (0.49–0.79) |
| Factual knowledge | 4.13 (2.49) | 3.2/2.0 | 0.89 (0.87–0.90) | 0.53 | 0.84 (0.73–0.90) |
| Accessing help | 3.34 (2.71) | 18.5/2.0 | 0.92 (0.91–0.93) | 0.69 | 0.72 (0.57–0.83) |
| Community support groups | 4.21 (2.89) | 13.1/6.2 | 0.92 (0.91–0.93) | 0.74 | 0.67 (0.50–0.79) |

Notes

[a] Mean scores are transformed to a range from 0–10

[b] $n = 503$; AIC = Average inter-item correlation; ICC = Intra-class correlation

[c] $n = 55$

## Discussion

In this mixed-methods study we used an iterative multi-step process of translation, adaptation, and validation that generated both qualitative and quantitative data for assessing several sources of validity evidence for the DLI-S with regard to culture, language, and care organization and provision. The results show evidence of cross-cultural validity of the DLI and confirm fit of the proposed six-factor model of DL. This supports other recent validation studies [7–9] as well as the potential of the DLI to be used to measure DL internationally.

Both I-CVI and S-CVI ratings exceed the recommended minimum set out by Polit et al. [30], suggesting support for DLI-S' validity in terms of item clarity and relevance for the DL construct, which aligns with Turkish results [8]. Despite high CVI ratings, several potential issues were raised by experts regarding item meaning and suitability in a Swedish context, highlighting the need for qualitative data to make meaningful assessments of content validity for translated items. Likewise, qualitative findings from the cognitive interviews were instrumental for guiding the researchers in addressing problematic items and unclear instructions. The cognitive interviews showed that the DLI-S could be completed by people with varying EOL experiences, from those with who describe themselves as having very limited EOL experiences to experts in the field. Importantly, the cognitive interviews also demonstrated that the DLI-S was not perceived as too sensitive or distressing, although it was noted that certain questions could bring up emotional memories. Several participants mentioned that they thought the items were interesting and thought-provoking, suggesting that completing the DLI-S could constitute a positive self-reflective experience. This is important since death education often emphasizes the role of reflection and sharing of experiences as part of the learning process [46]. In addition, the high proportion of survey participants who were willing to complete the

**Table 5. Fit indicators of tested factor models of death literacy ($n = 503$).**

| Tested model (Estimator DWLS) | CFI | TLI | RMSEA (CI) | SRMR | $\chi^2$/df |
|---|---|---|---|---|---|
| One factor | 0.933 | 0.928 | 0.200 (0.196–0.204) | 0.139 | 7966.767/377*** |
| Four factors | 0.980 | 0.978 | 0.110 (0.106–0.114) | 0.081 | 2629.432/371*** |
| Six factors | 0.993 | 0.993 | 0.064 (0.060–0.068) | 0.054 | 1107.631/362*** |

Notes: DLWS = Diagonal weighted least squares; TLI = Tucker-Lewis Index, CFI = Comparative Fit Index

***p <0.001

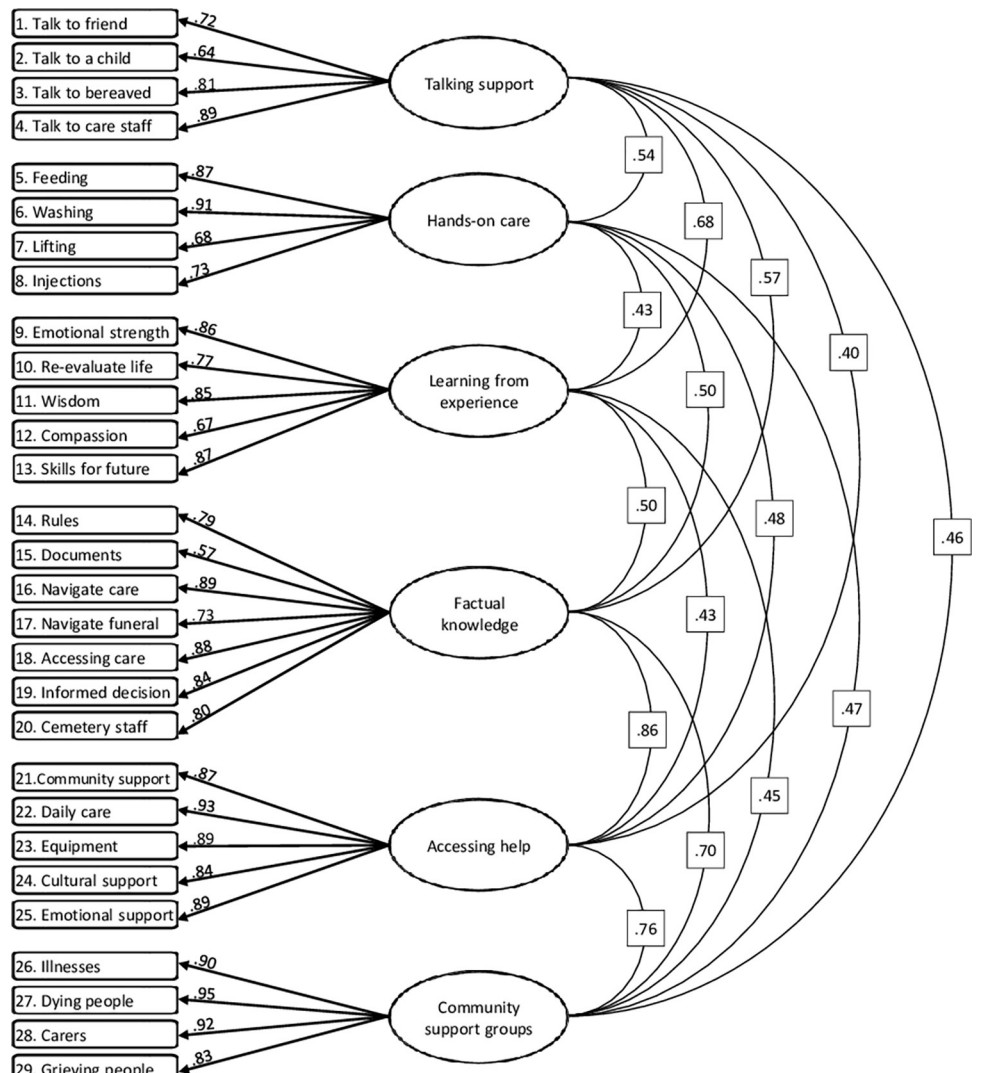

**Fig 2. Path diagram of the best-fitting model, demonstrating standardized factor loadings for items and correlations between factors.** Arrow Factor loadings. Line Correlation coefficients between factors.

survey a second time further illustrate DLI-S' acceptability. This is a significant finding since death is often described as a taboo topic in Sweden [47]. Still, as Arthur et al. [48] state, cognitive interviews about measures of a sensitive nature can be challenging: it may be difficult to raise concerns about intrusive or insensitive questions in a face-to-face situation, where a participant might feel obliged to justify their opinion.

One challenge regarding content validity concerns the definition of "community" used in item 21 (*Accessing help*). There is no Swedish word for "community", which could be translated with an emphasis on either social, geographical, or cultural connotations. To guide the translation process, the Swedish research team discussed the intention of the term with the original DLI developers. Based on this discussion, we used a translation that highlight location, i.e., neighborhood. Even if this is a common interpretation of the term and no major problems were identified during the cognitive interviews, it is possible that this translation was too narrow and influenced how the question functioned in the Swedish setting, as it is more specific than the English term.

The DLI-S was found to have satisfactory psychometric properties, with support for validity evidence based on internal structure, which corresponds to previous findings in validations of other language-versions of the DLI [2, 7–9]. However, our findings also identify some potential issues with the DLI-S that are important to consider. High Cronbach's α for all scales and sub-scales indicate that items are inter-related but does not necessarily mean that the scale is unidimensional [36]. Scale-level AICs further confirm strong item inter-relatedness, with values exceeding the recommended range. This finding raises questions of item redundancy, i.e., presence of items that do not sufficiently contribute with new information to measure the construct. The inter-item correlations suggest that the DLI-S might benefit from having one or several items removed: in particular the sub-scales comprising the dimensional scale *Community capacity* (*Accessing help* and *Community support groups*). These sub-scales consistently performed worse in comparison to the other scales and the DLI-S overall. For example, the floor effect in *Accessing help* indicates that the sub-scale has limited ability differentiating between responses at low levels, which might reduce reliability [12]. This finding also points to differences between the Swedish and Australian context that appear as variation in item difficulty for these items [49]. The distinctiveness of the Swedish context is further emphasized as the Community capacity scale functions well in China and Turkey [8, 9]. In addition, the confirmatory factor analysis showed that two factors (*Existing knowledge* and *Accessing help*) were highly correlated [41], suggesting that items in these scales may measure one, underlying, factor rather than two distinct dimensions of DL. Further studies are thus warranted to explore the performance of a shorter DLI-S version in the Swedish context and to investigate if a five-factor model constitutes a better fit for the DLI-S and the extent to which this might be relevant in other contexts. The Turkish and Chinese DLI validations similarly suggest that correlations between these two are the highest of the DLI scales, though not exceeding the cut-off off 0.85. This could indicate that knowledge about formal and informal systems and processes for care provision and support are not easily distinguished.

In sum, the validity evidence for internal structure show that the DLI-S performs well psychometrically, although the comparatively worse performance of the *Community capacity* sub-scales may indicate difference in function or meaning in Sweden compared to Australia, Turkey, China, and the United Kingdom. This finding could be an accurate reflection of differences in context, particularly in how care systems are organized and people's expectations of and interactions with them. In Sweden, public awareness of palliative and EOL care has been found to be generally low [46]. Unlike many other countries, Swedish palliative and EOL care is not dependent on public involvement such as volunteerism and charitable donations [50, 51]. Instead, Sweden has a long history of tax-funded universal welfare and high levels of trust in health care providers and institutions [52], which has remained stable even during the Covid-19 pandemic [53]. Indeed, participants in the cognitive interviews who gave low ratings for items in these sub-scales described feeling confident in their belief that if needed, they could turn to their primary care clinic for support or contact the national hub for information about health and healthcare services in Sweden that is accessible round-the-clock by phone or chat.

## Strengths and limitations

There are several methodological limitations that should be acknowledged. Participants in the online survey comprise a non-probability quota sample that was recruited from an existing national panel. Although the composition is balanced to that of the Swedish population in terms of age, gender, and place of residence; the educational level of our sample is slightly positively skewed compared to the population average [54] and underrepresented concerning place of birth, as 19.7% of the Swedish population are born outside Sweden, compared to only

7.2% of our participants [55]. More importantly, non-probability sampling raises concerns of possible self-selection and disproportion in unmeasured characteristics that may produce biased results, particularly if the purpose is making population estimates and representativeness [56, 57]. However, as theoretical rather than statistical generalization was the aim of this validation study, a representative and random sample was not required. Similarly, our 17% response rate may be considered low, but a high response rate is not necessary for the purpose of validation. Still, additional studies using larger and representative samples may allow further examination of the generalizability of the DLI-S' validity in the Swedish population. In addition, it was not possible to examine convergent validity in this study since Swedish translations of other validated measures of comparable constructs are lacking.

Despite these limitations, the study complied with recommended practice for translation, cultural adaptation, and validation, applying a rigorous process to assess validity and reliability from numerous sources of evidence [12, 18, 19]. In comparison to earlier DLI validations, we investigate the factor structure of DL in more detail, adding to the understanding of the fit between theoretical and factor model. It should be highlighted that since the aim of the study was to assess the validity of the DLI-S and not to develop a modified DLI version, no items were removed even if there were some indications of items that could be challenging in terms of comprehensibility (in phase 1) or may be redundant for measuring a DL dimension (in phase 2). Using bilingual field researchers instead of a professional translator during forward translation can be considered a strength, as proposed by Nolte et al. [58] who point out that professional translators often focus on the accuracy of the linguistic translation rather than general readability and conceptual meaning. We also made efforts to address previously identified issues of transparency in validation, e.g., providing full instructions for the expert panel review (S3 Appendix) to increase clarity regarding the basis of ratings [22]. An additional strength in this study is the use of literacy reviews, which is imperative for identifying possible unfair and unintended advantages or disadvantages to certain groups in the target population that might otherwise affect a measure's usefulness [59].

## Implications

Rather than measuring knowledge and skills alone, DL seems to represent a more overarching familiarity with the dying process, as recently suggested by Hayes et al. [60], that also encompasses attitudes and self-efficacy. This perspective seems fitting in the Swedish context, as our findings suggest that alongside gauging the extent of knowledge gained from prior EOL experiences, the DLI appears to capture perceived capacity to handle EOL-related issues and confidence in abilities to learn. The demonstrated acceptability and satisfactory psychometric properties of the DLI-S suggest that it is suitable to measure DL in Sweden. Nevertheless, more research is needed to better understand the DL construct, particularly across cultures, and to determine whether the DLI is appropriate to capture individual and community capacity for EOL in a comparable manner. Furthermore, the suitability of the DLI as an evaluation tool for EOL-related educational initiatives, both within and outside formal care settings, needs to be examined in Sweden and elsewhere.

Lack of public awareness of EOL care and civic preparedness for engaging with issues related to death and dying has been identified as hindering people's access to high-quality care [61]. Internationally, there is a growth of community based EOL interventions, such as compassionate communities, which are intended to encourage people to assist and support those at the EOL within their community. With further validation, the DLI has potential to be a multifaceted measure appropriate for continued cross-cultural research and better understanding of impact of such initiatives [62]. This is increasingly pertinent as it is expected that EOL care

provision will progressively take place outside formal care settings, e.g., aging populations, both internationally [63, 64] and in Sweden [65]. Additional research can also shed light on whether the DLI may be useful in the care context to measure overarching competence for EOL care among staff, especially to evaluate more integrated and comprehensive EOL education interventions [66, 67].

## Conclusion

This study provides empirical evidence supporting the validity of the Swedish translation and adaptation of the 29-item DLI to measure death literacy in the adult general public. The DLI-S was shown to be acceptable and feasible to answer regardless of the extent of respondents' prior EOL experiences. Our results support previous findings, indicating that the theoretical model operationalized in the DLI is stable across countries despite differences in language, culture, and organisation of institutions that all might influence local or regional death systems. In a time with growing interest in building community preparedness for EOL issues, the DLI-S constitutes a promising measure with good properties to capture overall capacity to engage with EOL care. Even though the six-factor model of the DLI yielded a good fit, our results show some characteristics that could potentially impact its measurement properties in a Swedish context.

## Supporting information

**S1 Appendix. Original English-language Death Literacy Index, with scale and sub-scale headings.**
(DOCX)

**S2 Appendix. Online survey, comprising the Swedish Death Literacy Index items and sociodemographic questions (in English).**
(DOCX)

**S3 Appendix. Instructions for expert panel review.**
(DOCX)

**S1 Table. Matrix with detailed examples of the revision process of two DLI-S items throughout instrument adaptation and validation.** Notes: DLI-S items 4 and 19, which were found to be problematic in terms of clarity, relevance, and/or language, are presented in Swedish and English. Item revision is marked in bold, with reasoning presented in English.
(DOCX)

**S2 Table. Ratings of relevance and clarity and calculated overall content validity index (CVI).** Notes: *Positive rating = number of experts rating the item 3 or 4 on a 4-point scale. I-CVI calculated as (*n* of raters rating 3 or 4/total *n* of raters). Scale-level CVI was calculated using average proportion of I-CVI values.
(DOCX)

**S3 Table. Inter-item correlation matrix of DLI-S items.** **. Correlation is significant at the 0.01 level (2-tailed). *. Correlation is significant at the 0.05 level (2-tailed).
(PDF)

## Acknowledgments

The authors would like to acknowledge the generous contributions by Aleksandra Sjöström-Bujacz, Leo Kowalski, and Sara Runesdotter to the statistical analyses. We also acknowledge the contribution of Terese Stenfors during the translation of the DLI-S.

## Author Contributions

**Conceptualization:** Therese Johansson, Carol Tishelman, Lars E. Eriksson, Ida Goliath, Joachim Cohen.

**Formal analysis:** Therese Johansson, Åsa Olsson, Carol Tishelman, Lars E. Eriksson, Ida Goliath, Joachim Cohen.

**Funding acquisition:** Ida Goliath.

**Investigation:** Therese Johansson, Åsa Olsson, Carol Tishelman, Lars E. Eriksson, Ida Goliath.

**Methodology:** Therese Johansson, Åsa Olsson, Carol Tishelman, Kerrie Noonan, Rosemary Leonard, Lars E. Eriksson, Ida Goliath, Joachim Cohen.

**Supervision:** Carol Tishelman, Lars E. Eriksson, Ida Goliath, Joachim Cohen.

**Writing – original draft:** Therese Johansson, Åsa Olsson.

**Writing – review & editing:** Therese Johansson, Åsa Olsson, Carol Tishelman, Kerrie Noonan, Rosemary Leonard, Lars E. Eriksson, Ida Goliath, Joachim Cohen.

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
