## [Decision Letter · Decision Letter 0]

26 Jun 2023

PONE-D-22-01686Validation of a culturally adapted Swedish-language version of the Death Literacy IndexPLOS ONE

Dear Dr. Johansson,

Thank you for submitting your manuscript to PLOS ONE. After careful consideration, we feel that it has merit but does not fully meet PLOS ONE’s publication criteria as it currently stands. Therefore, we invite you to submit a revised version of the manuscript that addresses the points raised during the review process.

We look forward to receiving your revised manuscript.

Kind regards,

Vilfredo De Pascalis

Academic Editor

PLOS ONE

“The authors would like to acknowledge the generous contributions by Aleksandra Sjöström-Bujacz, Leo Kowalski, and Sara Runesdotter to the statistical analyses. We also acknowledge the contribution of Terese Stenfors during the translation of the DLI-S.”

“This study was funded by Strategic Research Area Health Care Science (SFO‐V), Karolinska Institutet; the Doctoral School in Health Care Sciences, Karolinska Institutet; Swedish Research Council for Health, Welfare and Working Life (FORTE); Stockholm City Elder Care Bureau; and Stockholm Gerontology Research Center. The funders had no part in, nor influence on, the study design, data collection, analysis, interpretation, or writing of results.”

Additional Editor Comments:

Both reviewers consider the manuscript positively, although both suggested a revision; I invite the authors to solve all the raised issues and resubmit the revised version for publication acceptance.

Reviewers' comments:

Reviewer's Responses to Questions

**Comments to the Author**

1. Is the manuscript technically sound, and do the data support the conclusions?

Reviewer #1: Partly

Reviewer #2: Yes

2. Has the statistical analysis been performed appropriately and rigorously? 

Reviewer #1: Yes

Reviewer #2: Yes

3. Have the authors made all data underlying the findings in their manuscript fully available?

Reviewer #1: No

Reviewer #2: No

4. Is the manuscript presented in an intelligible fashion and written in standard English?

Reviewer #1: Yes

Reviewer #2: Yes

5. Review Comments to the Author

Reviewer #1: Hello

Thank you for writing the manuscript. It was well-done. But there are some issues in which to be clarify for publication:

1. The manuscript should be completed more. Reader need to know more about DL and DLI.

2. Materials and Methods: It is not complete because the process of Beaton et al. has not been completed. Why?

3. Materials and Methods: study design is not correct.

4. Usually Survey methods for harmonization of a scale or test are not valid.

5. How did you control inclusion and exclusion criteria for participants?

6. There are not inclusion and exclusion criteria in the text.

7. I didn't find psychometric parameters of the original scale, and authors didn't compare their data with it too.

8. In discussion, authors don't have any comparison perspective for their data. It is the base for discussion section.

Reviewer #2: The study translates and culturally adapts DLI into Swedish language and assess its validity. It is an interesting construct and professionally written. Few comments for the authors.

Abstract

i. Please provide the content validity index and fit indexes value(s) in the abstract.

Methods

i. Can you provide one or two sentences about the validity of the original DLI.

ii. Is it possible to merge all phases and sub-phases' data analysis together?

iii. It will be good to perform exploratory factor analysis in addition to confirmatory factor analysis. This will tell the reader the best model supported by the current data.

Results

i. Can you move 'scale descriptives' up? i.e. before 'reliability' results.

Discussion

i. Can you change the sub-title 'Methodological discussion' to 'strength and limitation'?

See other comments in the attached.

6. PLOS authors have the option to publish the peer review history of their article (what does this mean?). If published, this will include your full peer review and any attached files.

Reviewer #1: No

Reviewer #2: No

---

## [Author Response · Author response to Decision Letter 0]

30 Aug 2023

Reviewer #1: We thank the reviewer for taking the time to read our submitted manuscript and for their helpful comments and suggestions. Changes in response to feedback have been carefully considered. Please see our specific responses in the 'Response to reviewers' document.

Reviewer #2: We thank the reviewer for taking the time to read our submitted manuscript and for their helpful comments and suggestions. Changes in response to feedback have been carefully considered. Please see our specific responses in the 'Response to reviewers' document.

---

## [Decision Letter · Decision Letter 1]

2 Oct 2023

PONE-D-22-01686R1Validation of a culturally adapted Swedish-language version of the Death Literacy IndexPLOS ONE

Dear Dr. Johansson,

Thank you for submitting your manuscript to PLOS ONE. After careful consideration, we feel that it has merit but does not fully meet PLOS ONE’s publication criteria as it currently stands. Therefore, we invite you to submit a revised version of the manuscript that addresses the points raised during the review process.

Reviewers and myself are generally glad of the revised version of the manuscript. However, both reviewers raised some minor issues that must be addressed before the acceptance.

Thus, I invite the authors to make the suggested minor changes and resubmit their paper for the final decision.

We look forward to receiving your revised manuscript.

Kind regards,

Vilfredo De Pascalis

Academic Editor

PLOS ONE

Journal Requirements:

Additional Editor Comments:

Reviewers and myself are generally glad of the revised version of the manuscript. However, both reviewers raised some minor issues that must be addressed before the acceptance.

Thus, I invite the authors to make the suggested minor changes and resubmit their paper for the final decision.

Reviewers' comments:

Reviewer's Responses to Questions

**Comments to the Author**

1. If the authors have adequately addressed your comments raised in a previous round of review and you feel that this manuscript is now acceptable for publication, you may indicate that here to bypass the “Comments to the Author” section, enter your conflict of interest statement in the “Confidential to Editor” section, and submit your "Accept" recommendation.

Reviewer #1: (No Response)

Reviewer #2: (No Response)

2. Is the manuscript technically sound, and do the data support the conclusions?

Reviewer #1: Yes

Reviewer #2: Yes

3. Has the statistical analysis been performed appropriately and rigorously? 

Reviewer #1: Yes

Reviewer #2: Yes

4. Have the authors made all data underlying the findings in their manuscript fully available?

Reviewer #1: Yes

Reviewer #2: Yes

5. Is the manuscript presented in an intelligible fashion and written in standard English?

Reviewer #1: Yes

Reviewer #2: Yes

6. Review Comments to the Author

Reviewer #1: Thank you to let me to review the manuscript. The manuscript has been written well and all psychometric parameters have been considered. Results are acceptable.

My comments are:

1. Introduction: Although you have mentioned about other language versions of the scale, but there are not any data about them. The are necessary

2. Methods:

. Survey panel characteristics are not enough. It needs to be explained more.

. The method of translation and adaptation should be more precisely. Authors have pointed to multi-step mixed methods approach, but without any explanation about it.

3. Results:

. Authors should explain why they didn't do exploratory factor analysis.

4. discussion:

. There isn't any comparison between results of the present study and other language versions.

Reviewer #2: The manuscript has improved. There are two discrepancies noticed in the abstract, text body, table 4 and figure 2 which have not been attended to.

1. The sub scale alpha ranges between 0.81 and 0.92. Please correct it in the abstract and results sections (0.81 - 0.93).

2. The correlations between factors range between 0.40 and 0.86 in figure 2. Please correct it under results section (0.43 - 0.86).

7. PLOS authors have the option to publish the peer review history of their article (what does this mean?). If published, this will include your full peer review and any attached files.

Reviewer #1: No

Reviewer #2: No

---

## [Author Response · Author response to Decision Letter 1]

12 Nov 2023

Reviwer 1 and 2: We thank the reviewers for reading our resubmitted manuscript and for their constructive comments which were helpful in revising the manuscript. We have carefully considered and responded to each suggestion in the attached point-by-point response matrix in the Response to reviewers document.

---

## [Editor Report · Decision Letter 2]

16 Nov 2023

Validation of a culturally adapted Swedish-language version of the Death Literacy Index

PONE-D-22-01686R2

Dear Dr. Johansson,

We’re pleased to inform you that your manuscript has been judged scientifically suitable for publication and will be formally accepted for publication once it meets all outstanding technical requirements.

Kind regards,

Vilfredo De Pascalis

Academic Editor

PLOS ONE

Additional Editor Comments (optional):

The authors have made adequately all the suggested changes. Thus, I think that the manuscript can be accepted for publication.

I am sorry that the revision process took a long time. We finally reached a favorable decision.

My congratulations to the authors.
---

## [Editor Report · Acceptance letter]

21 Nov 2023

PONE-D-22-01686R2 

Validation of a culturally adapted Swedish-language version of the Death Literacy Index 

Dear Dr. Johansson:

I'm pleased to inform you that your manuscript has been deemed suitable for publication in PLOS ONE. Congratulations! Your manuscript is now with our production department. 

Kind regards, 

on behalf of

Prof. Vilfredo De Pascalis 

Academic Editor

PLOS ONE